# Healthcare Utilization and Costs after Receiving a Positive *BRCA1/2* Result from a Genomic Screening Program

**DOI:** 10.3390/jpm10010007

**Published:** 2020-02-03

**Authors:** Jing Hao, Dina Hassen, Kandamurugu Manickam, Michael F. Murray, Dustin N. Hartzel, Yirui Hu, Kunpeng Liu, Alanna Kulchak Rahm, Marc S. Williams, Amanda Lazzeri, Adam Buchanan, Amy Sturm, Susan R. Snyder

**Affiliations:** 1Department of Population Health Sciences, Geisinger, Danville, PA 17822, USA; dahassen@geisinger.edu (D.H.); yhu1@geisinger.edu (Y.H.); 2Division of Genetic and Genomic Medicine, Nationwide Children’s Hospital, Columbus, OH 43205, USA; Murugu.Manickam@nationwidechildrens.org; 3Department of Genetics, Yale School of Medicine, New Haven, CT 06510, USA; michael.murray@yale.edu; 4Phenomic Analytics and Clinical Data Core, Geisinger, Danville, PA 17822, USA; dnhartzel@geisinger.edu; 5Department of Computer Science, University of Central Florida, Orlando, FL 32816, USA; kunpengliu0827@gmail.com; 6Genomic Medicine Institute, Geisinger, Danville, PA 17822, USA; akrahm@geisinger.edu (A.K.R.); mswilliams1@geisinger.edu (M.S.W.); allazzeri@geisinger.edu (A.L.); ahbuchanan@geisinger.edu (A.B.); asturm@geisinger.edu (A.S.); 7Department of Health Policy and Behavioral Science, School of Public Health, Georgia State University, Atlanta, GA 30302, USA; ssnyder2@gsu.edu

**Keywords:** genomic screening, *BRCA1/2*, healthcare utilization, healthcare costs, uptake of risk management

## Abstract

Population genomic screening has been demonstrated to detect at-risk individuals who would not be clinically identified otherwise. However, there are concerns about the increased utilization of unnecessary services and the associated increase in costs. The objectives of this study are twofold: (1) determine whether there is a difference in healthcare utilization and costs following disclosure of a pathogenic/likely pathogenic (P/LP) *BRCA1/2* variant via a genomic screening program, and (2) measure the post-disclosure uptake of National Comprehensive Cancer Network (NCCN) guideline-recommended risk management. We retrospectively reviewed electronic health record (EHR) and billing data from a female population of *BRCA1/2* P/LP variant carriers without a personal history of breast or ovarian cancer enrolled in Geisinger’s MyCode genomic screening program with at least a one-year post-disclosure observation period. We identified 59 women for the study cohort out of 50,726 MyCode participants. We found no statistically significant differences in inpatient and outpatient utilization and average total costs between one-year pre- and one-year post-disclosure periods ($18,821 vs. $19,359, *p* = 0.76). During the first year post-disclosure, 49.2% of women had a genetic counseling visit, 45.8% had a mammography and 32.2% had an MRI. The uptake of mastectomy and oophorectomy was 3.5% and 11.8%, respectively, and 5% of patients received chemoprevention.

## 1. Introduction

Medical history-based *BRCA1/2* screening has been in place for two decades, offering testing based on personal and family history criteria. Recent data suggests that less than half of the individuals with *BRCA1/2* risk are identified through this approach [1]. Failure to identify cases are attributable to both the insensitivity of the approach and failure to universally apply screening [1,2]. Genomic screening programs in the general population for primary prevention is an alternative to identify individuals at risk for developing diseases who can benefit from enhanced preventive services, providing an intervention opportunity to reduce mortality and morbidity attributable to the screened conditions. There is increasing interest in implementing genomic screening programs for the general population as part of precision health, as evidenced by the All of Us research program and initiatives [3]. Genomic screening is likely to focus on the CDC tier 1 genomic applications [4,5], which are for conditions that have relatively higher prevalence, strong evidence for a gene–disease relationship, high penetrance with the potential for serious health impacts, and evidence-based effective interventions available for improving population health [6]. These include hereditary breast and ovarian cancer syndrome (HBOC), Lynch syndrome, associated with increased risk of colorectal, endometrial, and other cancers; and familial hypercholesterolemia, associated with increased risk of premature heart disease. Genetic testing is currently recommended for all these conditions in people at high risk (e.g., due to family history) [5,7]. An estimated two million people in the US have one of these conditions. Currently, these conditions are poorly ascertained by the healthcare system, and many individuals and their family members are not aware that they are at risk [8,9]. This has recently been shown in HBOC in two diverse settings [1,10]. It has been postulated that early detection through population genomic screening and intervention could potentially improve identification of risk and significantly reduce morbidity and mortality [5]. 

A central tenet of genomic screening in the general population is to proactively identify and manage risk at a reasonable cost [11]. However, while offering the potential to improve public health outcomes, it is unknown whether reporting population-based genomic screening results is followed by reasonable and appropriate healthcare utilization and costs. This is compounded by patient uncertainty concerning non-diagnostic genomic results, including how to use the result in the absence of other risk factors [12]. Thus, there are concerns as to whether returning genomic results from population genomic screening without a clinical indication could promote excessive and inappropriate utilization of services with the associated increases in costs, thus reducing or eliminating the potential value [13].

This study focused on one of the most prevalent and actionable conditions—HBOC, for which there is the support that genomic screening programs may better identify patients at high risk [1,14] compared to current guideline-based methods used to identify *BRCA1*/*2* carriers [7,15,16]. Screening for pathogenic/likely pathogenic (P/LP) variants in HBOC susceptibility genes in unselected individuals can be used to assess risk and guide decision-making about surveillance (MRI/mammogram) and prophylaxis (surgical and chemopreventive) for prevention and early detection of breast and ovarian cancer [17]. To date, evidence has not been available using real-world data on healthcare utilization and costs for the follow-up period after disclosure of P/LP *BRCA1/2* results via genomic screening. The objectives of this study are to: (1) determine whether there is a difference in healthcare utilization and costs following results disclosure, and (2) estimate the uptake of risk management per National Comprehensive Cancer Network (NCCN) guidelines within the first year of having a P/LP result returned.

## 2. Materials and Methods

### 2.1. Study Population 

The study population was identified from the Geisinger MyCode® Community Health Initiative (MyCode) where adult volunteers from an integrated healthcare delivery system in central Pennsylvania underwent exome sequencing using research protocols detailed in previous publications [1,18,19,20]. The study cohort was based on a population of 50,726 MyCode participants whose exome sequences were complete, were analyzed to identify potential P/LP variants, and had clinical confirmation of the research result with a report to the EHR to disclose the results between 01/01/2014 and 03/01/2016 [20]. The initial study cohort was defined as participants who received results of P/LP variants in *BRCA1/2*. The following inclusion criteria were further applied: female, a record of at least a one-year follow-up period post results disclosure and of being in the Geisinger Health System for at least one year prior to disclosure of the result, and no personal history of breast or ovarian cancer (Figure 1).

### 2.2. Data 

Participants’ demographic characteristics and healthcare utilization were obtained from the system’s electronic health record (EHR) and included all clinical encounters (inpatient, outpatient, emergency department, and other encounter types). Cost data were derived from the health system’s Decision Support Systems (DSS) electronic billing data, which contains detailed information on actual total costs and revenues/payments received for all medical services and physician procedures that each patient receives. We used the actual total payment amount received to calculate the costs of the patients’ medical care utilization from an insurance payor perspective. Surgical procedures and imaging procedures were identified using Current Procedural Terminology (CPT) codes. Imaging codes were supplemented with system-specific EHR codes. Chemoprevention uptake was obtained using medication identifiers and ICD9/ICD10 diagnosis codes (Appendix A). Genetic counseling visits were tracked in a database maintained by the genomic screening and counseling program. Chart review and additional study data sources were used to determine whether participants had prior clinical *BRCA1/2* genetic testing and to assess family history information for risk evaluation. In this study, a positive family history is defined as at least one first-degree relative or at least two second-degree relatives with *BRCA1/2*-related cancer (breast cancer, ovarian cancer, prostate cancer, pancreatic cancer, or melanoma). This information was based on a three-generation pedigree taken by a genetic counseling visit, a note from a provider, or the family medical history tab in the EHR regarding *BRCA1/2*-relevant cancers.

### 2.3. Study Design 

#### 2.3.1. Healthcare Utilization and Costs 

To evaluate whether there was a difference in healthcare service utilization and associated costs following the disclosure of a *BRCA1/2* result, all-cause healthcare utilization, i.e., the use of healthcare services for all reasons, and total costs, which included those associated with healthcare services of all encounter types in the year prior to the results’ disclosure, were compared to the year following the disclosure. We conducted this analysis for the entire cohort and then for a sub-cohort of patients who had previously not known their P/LP status.

The primary healthcare utilization outcome was outpatient visits to address a concern of patient over-utilization of healthcare services following the result disclosure (i.e., compared to other patient encounter types), because most of the recommended interventions are expected to occur in the outpatient setting. However, we also report inpatient encounters as surgical interventions such as mastectomy are inpatient procedures. We measured the percentage of women with any outpatient encounter, inpatient encounter, and the average per patient encounters by type. We analyzed the average total costs per patient in the pre- and post-disclosure periods. 

#### 2.3.2. Uptake of Guidelines-Recommended Risk Management

We analyzed the uptake of NCCN-recommended risk management in the first year post-disclosure. This includes surveillance recommendations using mammography and breast Magnetic Resonance Imaging (MRI), and recommendation or consideration of risk-reducing procedures and medications (mastectomy, salpingo-oophorectomy, chemoprevention). We also analyzed the uptake of genetic counseling visits as part of the MyCode genomic screening program. For uptake analysis, we conducted analysis only on patients who were eligible for the specific recommended service, e.g., for the uptake of mammogram/MRI, we only included women who were older than 25 years old [21]. The analysis of uptake of NCCN-recommended services was further stratified by age (≥40 years old, <40 years old), *BRCA1* and *BRCA2*, family history (positive, no, missing), and whether P/LP *BRCA1/2* was previously known. 

### 2.4. Statistical Analysis 

Descriptive statistics were used to summarize patient demographics, healthcare utilization, and costs in the pre- and post-disclosure periods, as well as the uptake of guideline-recommended risk management in the first year following the result disclosure. Statistical differences in healthcare utilization from pre- to post-disclosure were evaluated with McNemar’s exact test for categorical variables. We applied a boot-strapping approach to calculate 95 percent confidence intervals for continuous variables and corresponding *p*-values as the power calculation showed an insufficient sample size to conduct paired *t*-tests. For cost outcomes, we reported average and median total costs per patient and the average cost difference per patient in the pre- and post-disclosure periods, which were compared using the Wilcoxon signed-rank test due to the paired nature of the data. No discount rate was applied to the costs expressed in 2016–2017 US dollars given the relatively short 2-year study period and close proximity of the two time periods. All statistical analyses were conducted using Stata version 14 (StataCorp, College Station, TX, USA).

## 3. Results 

### 3.1. Patient Characteristics

The final study cohort included 59 women who received P/LP *BRCA1/2* results. Table 1 shows the demographics of the study cohort. The population was predominately white, not Hispanic or Latino. The mean age was 53 years, ranging from 24 to 90 years, with over 70% of patients being 40 years or older. The majority of the study population had the *BRCA2* variant (80%). There were 48 patients (81.4%) who had not undergone prior clinical testing and were unaware of their P/LP status. Family history of *BRCA1/2* relevant cancer was reported in 38 (64.4%) patients, while 12 (20.3%) patients reported a negative family history, and family history information was missing on the remaining 9 (15.3%) patients.

### 3.2. Healthcare Utilization 

Table 2 compares the healthcare utilization pre- and post-disclosure. In the entire study cohort, from the pre- to the post-disclosure period, there were no statistically significant changes observed in the percentage of patients with any outpatient (89.8% vs. 88.1%), inpatient (40.7% vs. 33.9%), or other encounter visits (74.6% vs. 74.6%). In terms of the average number of visits per patient, both the number of outpatient (7.37 vs. 6.93) and inpatient visits (0.63 vs. 0.51) did not change significantly from pre- to post-disclosure. A sub-analysis on those who had a previously unknown P/LP *BRCA1/2* variant (*n* = 48) showed similar results.

### 3.3. Costs 

The average total costs per patient were $18,821 pre-disclosure and $19,359 post-disclosure. The average difference ($538) observed in the post-disclosure period compared to the pre-disclosure period was not statistically significant (Table 3). A sub-analysis on those who had a previously unknown P/LP *BRCA1/2* variant (*n* = 48) showed a similar trend (Table 3). 

### 3.4. Uptake of Guidelines-Recommended Risk Management

During the first year post-disclosure, 29 (49.2%) women had a genetic counseling visit. About one-third (32.2%) of women in the cohort had an MRI and 45.8% had undergone mammography. Due to the one-year post-disclosure study timeframe coupled with the practice that some providers may apply of alternating breast MRI and mammogram every six months (“Breast Cancer Screening Guidelines. Memorial Sloan Kettering Cancer Center”, 2019), we included uptake of either a breast MRI or mammogram (49.2%). The overall uptake of mastectomy was 3.5% and uptake of oophorectomy was 11.8%. We observed uptake of chemoprevention at 5% (Table 4). The uptake rates in stratified sub-cohorts are detailed in Table 4.

## 4. Discussion 

This study provides real-world evidence to directly address the concern that genomic screening may lead to excessive and unnecessary increases in healthcare service utilization and costs [22,23]. The results of this study have not identified any statistically significant increase in overall use of healthcare services and associated costs even in a sub-cohort of patients who had previously unknown P/LP status and were assumed to be likely to have higher utilization and costs post-disclosure. The results also demonstrate the use of guideline-recommended preventive services. It is important to emphasize that these results are from a structured genomic screening program reporting results to patients with support offered from genetic counseling as part of the results disclosure process [18,20]. It is also noteworthy that the initial genetic counseling visits were considered part of the research study and thus were not charged; insurance and/or the patient covered the cost of any subsequent care. In practice, however, the initial genetic counseling visit might not be free of charge. 

The results of insignificance in healthcare utilization and cost differences are consistent with findings from the one other study identified providing similar evidence based on genome or exome sequencing and return of secondary findings (P/LP variants with associated phenotypes unrelated to the test indication), which concluded it added only modestly to near-term healthcare costs [24]. In the study by Hart et al., for 21 participants with P/LP *BRCA1/2* positive results, the average cost of follow-up medical actions per finding from a payer perspective for up to a one-year period was $115 and assuming all patients followed recommended actions $576 (2017 US dollars) based on nationally representative Medicare reimbursement rates, published estimates, wage data, and commercial lab sources to assign costs [24].

Our results on uptake of mastectomy (3.5%) and oophorectomy (11.8%) during the first year post-genomic screening in MyCode population was approximately a third of what is reported in published literature (uptake of mastectomy and oophorectomy about 10% and 30%–45%, respectively) [25,26,27] based on high-risk unaffected female *BRCA*1/2 carriers in the first year post-genetic testing. We conclude that the uptake of risk-reducing surgery may be lower in women ascertained through genomic screening among the general population versus a high-risk population based on family history. However, studying uptake at the population level is challenging because of differences in risk profiles, e.g., age, family history, socioeconomic status, and other population characteristics. 

The major strength of this study is the results are based on real-world data of unaffected patients in a genomic screening program with the return of P/LP *BRCA1/2* positive results. To date, information of this type has not been available in other published studies that either have relied on high-risk populations [26,27,28] and/or imputed or estimated values in economic evaluation models [29]. This study has several limitations. The study population is composed of volunteers within a single state and a limited geographic area (central and northeastern Pennsylvania). The study population is relatively small (*n* = 59), although this is a common feature in previous studies of *BRCA1/2* positive populations, and is not broadly representative as it contains larger proportions of key subgroups (older age, non-Hispanic European ancestry, and family history of breast and ovarian cancer). Thus, the results cannot be generalized to other populations. The study results encompass healthcare services obtained within a single healthcare system (Geisinger), although individuals in the study population may have used healthcare services outside the system which were not included in the study data sources. Geisinger is the largest healthcare system in central Pennsylvania and all recommended services are provided within the Geisinger system so participants did not have to leave the system to obtain these services. The study’s post-disclosure time period was limited to one year, which may have resulted in lower estimates for both research questions concerning overall healthcare utilization, costs and guideline-recommended services. Lastly, due to the nature of the billing data source for costs, the costs of outpatient visits, the most relevant for this study, could not be analyzed separately.

An important role of *BRCA1/2* testing is to identify high-risk variant carriers before they develop breast or ovarian cancer, so they may start cancer screening and preventive interventions to reduce cancer risk. However, evidence shows that current clinical genetic testing practice, which is primarily based on family history criteria and a predefined mutation probability threshold, is inefficient, resource-intensive, and misses >50% of individuals or mutation carriers at risk [1,14,30]. In a 2018 systematic review of population-based testing, the authors concluded that large scale general population-based screening can overcome these limitations as genetic-testing costs are falling and acceptability and awareness are increasing, yet there are few large-scale programs for unaffected women and a lack of evidence for their costs and consequences [14]. Despite the positive trend in rates of *BRCA1/2* testing, [31] the majority of at-risk women do not get a referral for genetic counseling and testing, and a large majority of *BRCA1/2* variant carriers in the US have not been identified [32]. Consequently, genomic screening has been suggested for the general population to prevent hereditary breast and ovarian cancers [8,33]. This study attempts to address the concern that population-based genomic screening programs may lead to an unnecessary increase in healthcare service utilization and costs, particularly in comparison to criteria-based genetic testing programs targeting those at higher risk, and offers initial evidence to mitigate such concerns. 

While population genomic screening for *BRCA1/2* offers the potential to transform public health by preventing disease and saving lives, there are challenges to implementing such programs that constrain the possibility of widespread adoption despite declining sequencing/testing costs. There remains a substantial need for evidence development of population genomic screening [19] including: how to and who should provide testing and the recommended health services for those at high risk to ensure quality and equitable access to services; whether it causes more good than harm and can provide good value in terms of cost-effectiveness or other measures; and whether and under what conditions is there consumer demand. In addition, there are complex ethical, legal, and social issues including privacy and discrimination as well as cultural, religious, psychological, and economic factors. 

Future studies within integrated healthcare systems and including larger and more diverse study populations and longer follow-up periods can provide further evidence of the medical care, costs, and health outcomes of *BRCA1/2* positive individuals identified by genomic screening programs. Rigorous and context-specific models are needed to assess a range of possible scenarios in which a broader population genomic screening might have added value. Careful assessment of the benefits, risks, and cost-effectiveness of population screening is needed before the introduction of costly large-scale interventions, as published evidence is sparse on the potential effects on individuals and the healthcare system. This additional evidence is an initial step that informs the implementation of genomic medicine and enables the timely realization of its benefits.

## Figures and Tables

**Figure 1 jpm-10-00007-f001:**
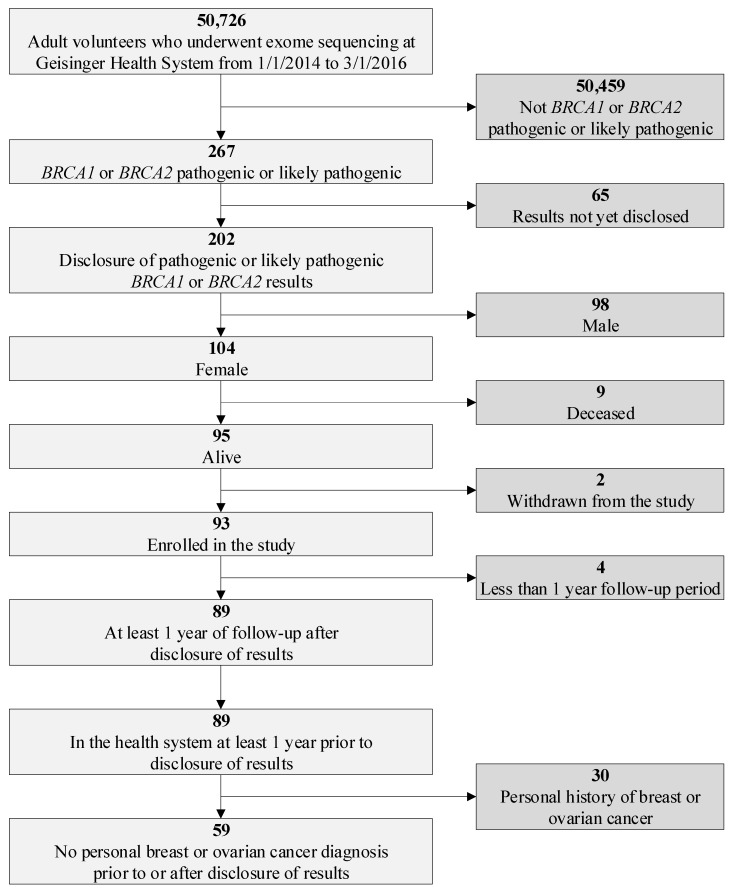
Study population.

**Table 1 jpm-10-00007-t001:** Demographic characteristics.

	N	Percentage
Total Study Population	59	
*BRCA* Variant		
*BRCA1*	12	20.3%
*BRCA2*	47	79.7%
Age (Mean), (Min, Max) Years	53	24,90
Age < 40 years old	16	27.1%
Age ≥ 40 years old	43	72.9%
Race		
Black or African American	1	1.7%
White	58	98.3%
Ethnicity		
Hispanic or Latino	2	3.4%
Not Hispanic or Latino	57	96.6%
Previously Known	11	18.6%
Previously Unknown	48	81.4%
Prior Mastectomy	2	3.4%
No Prior Mastectomy	57	96.6%
Prior Oophorectomy	8	13.6%
No Prior Oophorectomy	51	86.4%
Family History of *BRCA* Relevant Cancer		
With Family History of *BRCA* Relevant Cancer	38	64.4%
Without Family History of *BRCA* Relevant Cancer	12	20.3%
Missing Family History Information	9	15.3%

**Table 2 jpm-10-00007-t002:** Healthcare utilization in the pre- and post-disclosure of results.

	**Pre-Disclosure**	**Post-Disclosure**	***P*-Value**	**Average Difference Per Patient (Post—Pre-Disclosure)**	**Boot-Strapping 95% CI**	**Boot-Strapping *p*-Value**
Patient with any outpatient visits, No. (%)	53 (89.83)	52 (88.14)	1.00 ^a^			
Patients with any inpatient admissions, No. (%)	24 (40.68)	20 (33.90)	0.56 ^a^			
Patient with any other encounter visits, No. (%)	44 (74.58)	44 (74.58)	1.00 ^a^			
No. of Outpatient visits per patient, Mean (SD)	7.37 (6.25)	6.93 (5.34)	0.52	−0.44	(−1.76, 0.88)	0.51
No. Inpatient encounters per patient, mean (SD)	0.63 (1.02)	0.51 (0.86)	0.49	−0.12	(−0.46, 0.22)	0.49
**Sub-cohort of Previously Unknown (*N* = 48)**	**Pre-Disclosure**	**Post-Disclosure**	***P*-Value**	**Average Difference Per Patient (Post—Pre-Disclosure)**	**Boot-Strapping 95% CI**	**Boot-Strapping *p*-Value**
Patient with any outpatient visits, No. (%)	43 (89.58)	41 (85.42)	0.73 ^a^			
Patients with any inpatient admissions, No. (%)	18 (37.50)	15 (31.25)	0.66 ^a^			
Patient with any other encounter visits, No. (%)	35 (72.92)	34 (70.83)	1.00 ^a^			
No. of Outpatient visits per patient, Mean (SD)	7.13 (6.54)	6.92 (5.57)	0.79	−0.21	(−1.72, 1.30)	0.79
No. Inpatient encounters per patient, mean (SD)	0.58 (1.03)	0.48 (0.85)	0.61	−0.10	(−0.49, 0.28)	0.59

^a^*p*-values based on McNemar’s exact test. CI, confidence interval.

**Table 3 jpm-10-00007-t003:** Cost in the pre- and post-disclosure of results.

**Entire Cohort (*N* = 59)**	**Pre-Disclosure**			**Post-Disclosure**		
	**Average**	**Median**	**Interquartile Range**	**Average**	**Median**	**Interquartile Range**
Total costs per patient ($)	$18,821	$4956	($1240, $19,318)	$19,359	$4,622	($1084, $27,738)
**Average Difference Per Patient**	$538 (*p* = 0.76 ^a^)					
	**Pre-Disclosure**			**Post-Disclosure**		
**Patients Previously Unknown (*N* = 48)**	**Average**	**Median**	**Interquartile Range**	**Average**	**Median**	**Interquartile Range**
Total costs per patient ($)	$15,122	$4197	($1157, $17,381)	$17,699	$4955	($1062, $22,906)
**Average Difference Per Patient**	$2578 (*p* = 0.76 ^a^)					

^a^*p*-value based on the Wilcoxon signed-rank sum test.

**Table 4 jpm-10-00007-t004:** Uptake of guidelines-recommended services within the first year post results disclosure.

(No., %)	N	N(MAST)	N(OOPH)	N(MAMMO/MRI)	MAST	OOPH	MAMMO	MRI	MAMMO or MRI	CHEMO	GC
Overall	59	57	51	58	2 (3.5)	6 (11.8)	27 (46.6)	19 (32.8)	29 (50.0)	3 (5.1)	29 (49.2)
Age <40	16	15	13	15	1 (6.7)	1 (7.7)	6 (40.0)	5 (33.3)	7 (46.7)	1 (6.3)	6 (37.5)
Age ≥40	43	42	38	43	1 (2.4)	5 (13.2)	21 (48.8)	14 (32.6)	22 (51.2)	2 (4.7)	23 (53.5)
BRCA1	12	10	8	12	0 (0)	1 (12.5)	5 (41.7)	4 (33.3)	6 (50.0)	1 (8.3)	1 (8.3)
BRCA2	47	47	43	46	2 (4.3)	5 (11.6)	22 (47.8)	15 (32.6)	23 (50.0)	2 (4.3)	28 (59.6)
Previously Known	11	9	8	11	0 (0)	2 (25)	5 (45.5)	5 (45.5)	6 (54.5)	2 (18.2)	1 (9.1)
Previously Unknown	48	48	43	47	2 (4.2)	4 (9.3)	22 (46.8)	14 (29.8)	23 (48.9)	1 (2.1)	28 (58.3)
With Family History	38	36	32	38	2 (5.6)	4 (12.5)	21 (55.3)	15 (39.5)	23 (60.5)	1 (2.6)	20 (52.6)
With No Family History	12	12	11	11	0 (0)	2 (18.2)	6 (54.5)	4 (36.4)	6 (54.5)	2 (16.7)	9 (75.0)
Missing Family History	9	9	8	9	0 (0)	0 (0)	0 (0)	0 (0)	0 (0)	0 (0)	0 (0)

Note: CHEMO, chemoprevention medication; GC, genetic counseling visit; MAMMO, mammogram; MRI, (breast) Magnetic Resonance Imaging; MAST, mastectomy; No., number (count); OOPH, oophorectomy. For mastectomy uptake analysis, we excluded 2 individuals who had a prior mastectomy (overall *n* = 57); for oophorectomy uptake analysis, we excluded 8 individuals who had a prior oophorectomy (overall *n* = 51); for imaging (mammogram/MRI) uptake analysis, we excluded one woman who was younger than 25 years old because breast imaging recommendations begin starting at age 25 according to the NCCN guidelines [11] (overall *n* = 58). For all other uptake variables, (overall *n* = 59).

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
