# Peer review of "Healthcare Utilization and Costs after Receiving a Positive BRCA1/2 Result from a Genomic Screening Program"

_jpm, 2020, doi:10.3390/jpm10010007_

Round 1

Reviewer 1 Report

Page 7: Table 1: Add "Percentage" on the top row next to "N"

Page 5: 2.2 data: Elaborate on how cost was obtained from various sources. focus on the perspective of cost, was it insurance perspective? 

Suggest performing sensitivity analysis with 20% (+ or -) cost variation to demonstrate that the results would remain the same if the cost vary.

Add a limitation section which states that the sample was too small and the results cannot be generalized.

have you considered Chi squar test? I feel that would be the most appropriate test for analysis considering that your data cells are very asymmetric.

Author Response

Our response is copied here and is also included in the response letter Word Document.  

Page 7: Table 1: Add "Percentage" on the top row next to "N"

Response: We thank the reviewer for the careful review. As requested, the word “Percentage” is now added in Table 1 (page 7).  

Page 5: 2.2 data: Elaborate on how cost was obtained from various sources. focus on the perspective of cost, was it insurance perspective? 

Response: We appreciate this request from the reviewer as the issue of cost, how it is defined, and the perspective can vary, and this offers an opportunity to better clarify and define our use of costs consistent with the study data and approach used. In this study, the single data source for costs was electronic billing data (i.e. the Decision Support Systems (DSS) billing data at Geisinger) which contains detailed information of actual total internal accounting costs and actual revenues received/payments from payors for patient care including hospital services and physician procedures. In our analysis, we used the health system’s net revenue, more specifically, actual total payment amount received to calculate the costs of the patients’ medical care utilization from an insurance payor perspective.

Based on the reviewer’s comment, we realize that we could expand our statement in the manuscript Methods section describing cost and make the cost data source and perspective more explicit. We have revised our statement on pages 5-6 from:  “Cost data were derived from payment amounts obtained from electronic billing data.”

to:

“Cost data were derived from the health system’s Decision Support Systems (DSS) electronic billing data which contains detailed information of actual total costs and revenues/payments received for all hospital services and physician procedures that each patient receives. We used the actual total payment amount received to calculate the costs of the patients’ medical care utilization from an insurance payor perspective.”

Suggest performing sensitivity analysis with 20% (+ or -) cost variation to demonstrate that the results would remain the same if the cost vary.

Response: We appreciate the reviewer’s suggestion, and as it is common in health economic analyses, we tried to figure out how a sensitivity analysis on costs can be applied to this study. However, as defined, sensitivity analysis (SA) is ““a method to determine the robustness of an assessment by examining the extent to which results are affected by changes in methods, models, values of unmeasured variables, or assumptions” with the aim of identifying “results that are most dependent on questionable or unsupported assumptions”.  Essentially, SA addresses the “what-if-the-key-inputs-or-assumptions-changed”-type of question.”[1] SA including varying costs by some percentage is commonly used in health economics especially in cost-effectiveness modeling when we have uncertainties with cost values which may have a substantial impact on the study outcomes (commonly incremental cost effectiveness ratio, ICER) or there are outlier observations (i.e., exceptional cases in the sample).

In our study, we conducted an empirical data analysis on differences in patients’ utilization and costs pre- and post-disclosure of genomic screening results. The costs were derived from a single data source, i.e. the financial billing data of our healthcare system which reflects the real-world actual revenue/payment received by a healthcare system from payors for healthcare services that patients received. The main cost outcomes show that the difference in costs pre- and post-disclosure were not statistically significant. Sensitivity analysis is typically a re-analysis of either the same outcome using different approaches, or different definitions of the outcome—with the primary goal of assessing how these changes impact the conclusions. Varying the cost values by ±20% would not impact this outcome/conclusion as the costs would change in both pre- and post-periods. In fact, we did a sub-analysis in the sub-cohort of patients previously unknown of their pathogenic/likely pathogenic status assuming this would be the cohort that has higher utilization and costs post-disclosure, and the conclusion still held the same. Thus, we feel comfortable with our results/observation in our healthcare system. We have added a sentence in the discussion section (page 10) to highlight that the sub-cohort findings stay the same. Please let us know if you still have a concern or require further clarification for this comment/suggestion.

“The results of this study have not identified any statistically significant increase in overall use of healthcare services and costs even in a sub-cohort of patients who had previously unknown P/LP status and were assumed to be likely to have higher utilization and costs post-disclosure.”

Add a limitation section which states that the sample was too small and the results cannot be generalized.

Response: We thank the reviewer for pointing out this important limitation. We did acknowledge this limitation of small sample and generalizability in the discussion section:  “The study population is relatively small (n=59), (although this is a common feature in previous studies of BRCA1/2 positive populations), and is not broadly representative as it contains larger proportions of key subgroups (older age, non-Hispanic European ancestry, and family history of breast and ovarian cancer). This reduces the generalizability of this study to other populations.” However, we agree with the reviewer that we could be more explicit on the non-generalizability of the results, thus, we revised the statements (page 11):

“The study population is relatively small (n=59), (although this is a common feature in previous studies of BRCA1/2 positive populations), and is not broadly representative as it contains larger proportions of key subgroups (older age, non-Hispanic European ancestry, and family history of breast and ovarian cancer). Thus, the results cannot be generalized to other populations.” 

have you considered Chi squar test? I feel that would be the most appropriate test for analysis considering that your data cells are very asymmetric.

Response: We thank reviewer for suggesting this potential. We did carefully consider the options for test choices based on different types of data (please refer to our statistical analysis statements copied below). For categorical data which a Chi square test would be typically considered, due to the paired nature of the data, i.e. we are comparing the same patient in pre- and post-disclosure periods, we decided to use McNemar’s exact test which is designed to assesses the difference between paired proportions. 

“Statistical differences in healthcare utilization from pre- to post-disclosure were evaluated with McNemar’s exact test for categorical variables.  We applied a bootstrapping approach to calculate 95 percent confidence intervals for continuous variables and corresponding p-values as the power calculation showed insufficient sample size to conduct paired t-tests. For cost outcomes, we reported average and median total costs per patient and the average cost difference per patient in the pre- and post-disclosure periods, which are compared using the Wilcoxon signed rank test due to the paired nature of the data.”  

[1] A tutorial on sensitivity analyses in clinical trials: the what, why, when and how Lehana Thabane, Lawrence Mbuagbaw, Shiyuan Zhang, Zainab Samaan, Maura Marcucci, Chenglin Ye, Marroon Thabane, Lora Giangregorio, Brittany Dennis, Daisy Kosa, Victoria Borg Debono, Rejane Dillenburg, Vincent Fruci, Monica Bawor, Juneyoung Lee, George Wells, Charles H Goldsmith BMC Med Res Methodol. 2013; 13: 92.  Published online 2013 Jul 16. doi: 10.1186/1471-2288-13-92 PMCID: PMC3720188

Reviewer 2 Report

A thoroughly conducted study that assessed the Healthcare Utilization and Costs following two Positive tests for BRCA1/2  from a Genomic Screening Program. Authors need to justify why they did not conduct a sensitivity analysis to further characterize the robustness of their findings. 

Author Response

We copied the response here and also included in the response letter Word Document. 

A thoroughly conducted study that assessed the Healthcare Utilization and Costs following two Positive tests for BRCA1/2  from a Genomic Screening Program. Authors need to justify why they did not conduct a sensitivity analysis to further characterize the robustness of their findings. 

Response: We appreciate the reviewer’s positive review of our study!

As indicated in the response above, sensitivity analysis (SA) is ““a method to determine the robustness of an assessment by examining the extent to which results are affected by changes in methods, models, values of unmeasured variables, or assumptions” with the aim of identifying “results that are most dependent on questionable or unsupported assumptions”.  Essentially, SA addresses the “what-if-the-key-inputs-or-assumptions-changed”-type of question.”[1]

The main outcomes of interests in this study were the differences in patients’ healthcare utilization and costs in pre- and post-disclosure of results periods. The study is an observational study to examine the experience in one healthcare system utilizing relatively simple methods and analytical approaches (descriptive analysis and paired two group comparisons) based on real-world electronic health record data. Thus, there are limited uncertainties in the data, assumptions and methods for sensitivity analyses. Thinking of what would impact the outcomes, we did sub-analysis in the sub-cohort of patients with previously unknown pathogenic/likely pathogenic status assuming this would be the cohort with likely higher utilization and costs post-disclosure, and the same conclusions still held. Thus, we feel comfortable with our results/observation in our healthcare system. We have added a sentence in discussion (page 10) to highlight that the sub-cohort findings stay the same.

“The results of this study have not identified any statistically significant increase in overall use of healthcare services and costs even in a sub-cohort of patients who had previously unknown P/LP status and were assumed to be likely to have higher utilization and costs post-disclosure.”

[1] A tutorial on sensitivity analyses in clinical trials: the what, why, when and how Lehana Thabane, Lawrence Mbuagbaw, Shiyuan Zhang, Zainab Samaan, Maura Marcucci, Chenglin Ye, Marroon Thabane, Lora Giangregorio, Brittany Dennis, Daisy Kosa, Victoria Borg Debono, Rejane Dillenburg, Vincent Fruci, Monica Bawor, Juneyoung Lee, George Wells, Charles H Goldsmith BMC Med Res Methodol. 2013; 13: 92.  Published online 2013 Jul 16. doi: 10.1186/1471-2288-13-92 PMCID: PMC3720188